# Automation, live-cell imaging, and endpoint cell viability for prostate cancer drug screens

**Rolando D. Z. Lyles**[1,2], **Maria J. Martinez**[2,3], **Benjamin Sherman**[2,3], **Stephan Schürer**[2,3], **Kerry L. Burnstein**[2,3] *

1 Sheila and David Fuente Graduate Program in Cancer Biology, University of Miami Miller School of Medicine, Miami, Florida, United States of America, 2 Sylvester Comprehensive Cancer Center, University of Miami, Miami, Florida, United States of America, 3 Department of Molecular & Cellular Pharmacology, University of Miami Miller School of Medicine, Miami, Florida, United States of America

* kburnstein@med.miami.edu

**Data Availability Statement:** All relevant data are within the paper and its Supporting Information files.

**Funding:** Award Recipient: KLB Grant Number: 9BC13 Grant Title: Data-Driven Identification of

## Abstract

Androgen deprivation therapy (ADT) is the standard of care for high risk and advanced prostate cancer; however, disease progression from androgen-dependent prostate cancer (ADPC) to lethal and incurable castration-resistant prostate cancer (CRPC) and (in a substantial minority of cases) neuroendocrine prostate cancer (NEPC) is common. Identifying effective targeted therapies is challenging because of acquired resistance to established treatments and the vast heterogeneity of advanced prostate cancer (PC). To streamline the identification of potentially active prostate cancer therapeutics, we have developed an adaptable semi-automated protocol which optimizes cell growth and leverages automation to enhance robustness, reproducibility, and throughput while integrating live-cell imaging and endpoint viability assays to assess drug efficacy in vitro. In this study, culture conditions for 72-hr drug screens in 96-well plates were established for a large, representative panel of human prostate cell lines including: BPH-1 and RWPE-1 (non-tumorigenic), LNCaP and VCaP (ADPC), C4-2B and 22Rv1 (CRPC), DU 145 and PC3 (androgen receptor-null CRPC), and NCI-H660 (NEPC). The cell growth and 72-hr confluence for each cell line was optimized for real-time imaging and endpoint viability assays prior to screening for novel or repurposed drugs as proof of protocol validity. We demonstrated effectiveness and reliability of this pipeline through validation of the established finding that the first-in-class BET and CBP/p300 dual inhibitor EP-31670 is an effective compound in reducing ADPC and CRPC cell growth. In addition, we found that insulin-like growth factor-1 receptor (IGF-1R) inhibitor linsitinib is a potential pharmacological agent against highly lethal and drug-resistant NEPC NCI-H660 cells. This protocol can be employed across other cancer types and represents an adaptable strategy to optimize assay-specific cell growth conditions and simultaneously assess drug efficacy across multiple cell lines.

## Introduction

Androgen deprivation therapy (ADT) is the standard-of-care for high risk localized or advanced prostate cancer; however, disease progression from androgen-dependent (AD) to lethal and incurable castration -resistant prostate cancer (CRPC) and (in a substantial minority

Novel Precision Drug Combination Therapies for Prostate Cancer Sponsor: Florida Department of Health Program: William G. "Bill" Bankhead, Jr., and David Coley Cancer Research Program Website: https://www.floridahealth.gov/provider-and-partner-resources/research/funding-opportunity-announcements/coleyfoa.html No, the funders did not and will not have a role in study design, data collection and analysis, decision to publish, or preparation of the manuscript. Award Recipient: KLB Grant Number: No number assigned Grant Title: Targeting KIF20 a critical promoter of castration resistant prostate cancer in robust pre-clinical models Sponsor: The Mike Slive Foundation Program: Prostate Cancer Research Pilot Grant Program Website: https://mikeslivefoundation.org/ No, the funders did not and will not have a role in study design, data collection and analysis, decision to publish, or preparation of the manuscript. Award Recipient: KLB Grant Number: No number assigned Grant Title: Internal funds supporting Dr. Kerry Burnstein, Associate Director, Education and Training Sponsor: Sylvester Comprehensive Cancer Center at the University of Miami Program: Professorship Website: https://umiamihealth.org/sylvester-comprehensive-cancer-center/about-sylvester/leadership/leadership No, the funders did not and will not have a role in study design, data collection and analysis, decision to publish, or preparation of the manuscript.

**Competing interests:** The authors have declared that no competing interests exist.

of cases) neuroendocrine prostate cancer (NEPC) is common [1–6]. Treatments for patients with these highly heterogeneous cancers result in modest survival benefits highlighting the need for more robust and efficient systems to identify potential new therapeutics targets for prostate cancer (PC).

In vitro compound screening assays are the cornerstone to discovering potentially active agents in cancer research. There are three core steps to all effective therapeutic drug screening methods: 1) selecting the correct model system(s), 2) defining the compound/drug concentration range to be tested and treatment duration, and 3) measuring drug effectiveness via a quantifiable assay [7–9]. Even with similar in vitro models and experimental parameters (e.g., tested drugs, concentration ranges, treatment duration) it can be difficult to compare and reproduce results across labs due to the use of different methodologies to evaluate drug performance. Most drug screens measure cell viability to determine response. Popular techniques to assess cell viability include dye exclusion assays, colorimetric assays, nuclear staining, and hemocytometry, but the most widely used and easily scalable methods are endpoint metabolic assays and live-cell imaging. Endpoint viability assays such as the CellTiter-Glo or the (3-(4,5-dimethylthiazol-2-yl)-2,5-diphenyltetrazolium bromide (MTT) assays measure drug effectiveness based on cell metabolism as a proxy for live cells [10, 11]. Real-time microscopy platforms track cell morphology, proliferation, cell death and confluence in fluorescence-enhanced or label-free contexts [12, 13]. Ultimately, both endpoint viability assays and real-time imaging platforms have flaws [14–17] and data support using both methods in tandem to optimally analyze cellular drug response [15, 18].

Here we report the design of an adaptable semi-automated drug screening protocol which integrates live-cell imaging and endpoint viability to evaluate the efficacy of anti-cancer therapeutic compounds, repurpose drugs, and to provide a starting point for future analysis of drug combinations across a genetically and phenotypically diverse panel of prostate cells. This protocol has two arms: Arm 1) 96-well plate cell growth optimization for integrated live-cell and endpoint viability drug screening assay. and Arm 2) 96-well plate OT-2 liquid handler integrated live-cell and endpoint viability drug activity screen. This protocol incorporates automation via the Opentrons OT-2 liquid handler, real-time imaging via the IncuCyte ZOOM Live-cell Analyzer, and the CellTiter-Glo ATP-based endpoint viability assay. Our approach has three key benefits: 1) assay-specific cell growth optimization: allowing for simultaneous and comparative screening of a single compound across a range of cancer phenotypes; 2) automation: minimizing sources of error, increasing efficiency, and enhancing reproducibility during cell seeding and drug preparation/delivery; 3) integration of live-cell imaging and endpoint viability assays: providing visual representation of overall cell viability and integrity throughout experiments, endpoint drug performance metrics, identification of cytostatic vs cytotoxic effects of tested drugs, and internal controls limiting false negative classifications of tested compounds. The data produced in a single experimental run include real-time representation of morphological changes, cell confluence/growth rates, and quantifiable pharmacological performance metrics including (but not limited to) traditional dose-response curves, growth rate inhibition (GR) dose-response curves, IC50, and GR50. Although this study was designed for prostate cancer model systems, the protocol is highly adaptable and both optimization and drug assessment arms of the protocol can be used independently and for other cancer types.

## Materials & methods

The methods and protocol described in this peer-reviewed article are published on protocols. io (**dx.doi.org/10.17504/protocols.io.x54v9dojzg3e/v1**) and is included for printing as S1 File with this article.

## Cell lines

Prostate cell lines were obtained from the American Type Culture Collection (ATCC), and collections curated from investigators who developed select cell lines which have been maintained in our laboratory. In total, nine cell lines were characterized, authenticated (LabCorp), STR profiled, and mycoplasma tested for validation. The prostate cell lines used were: BPH-1 and RWPE-1 (non-tumorigenic), LNCaP and VCaP (androgen-dependent prostate cancer (ADPC)), C4-2B and 22Rv1 (castration-resistant prostate cancer (CRPC)), DU 145 and PC3 (androgen receptor-null CRPC), and NCI-H660 (neuroendocrine prostate cancer (NEPC)).

## Tissue culture

For maintenance, cells were cultured in their recommended media. BPH-1, LNCaP, 22Rv1, PC3 and DU 145 cells were cultured in RPMI (Corning, #15-040-CV) supplemented with 1% penicillin/streptomycin (ThermoFisher, #15140122), 1% L-glutamine (ThermoFisher, #25030024), and 10% FBS (Atlanta Biologicals). VCaP cells were cultured in DMEM Gluta-MAX (Gibco, #10569–044) supplemented with 1% Anti-Anti (ThermoFisher, #15240062), and 10% FBS. C4-2B cells were cultured in DMEM (Corning, #10-013-CV) supplemented with 1% penicillin/streptomycin, 1% L-glutamine, and 10% FBS. RWPE-1 cells were cultured in KSFM (ThermoFisher, #17005042) supplemented with 1% penicillin/streptomycin, 25mg Bovine Pituitary Extract (Gibco, #13028–014), and 2.5 µg human recombinant EGF (Gibco, #10450–013). NCI-H660 cells were cultured in RPMI-1640 High-Glucose (ATCC, #30–2001) supplemented with 1% penicillin/streptomycin, 2mM L-glutamine (additional), 1% Insulin Transferrin-Selenium (Gibco, 41400–045), 10nM Hydrocortisone (Sigma-Aldrich, #H0135), 10nM β-estradiol (Sigma-Aldrich, #E2257), and 5% FBS. During the drug screening assay all CRPC (C4-2B, PC3, DU 145, 22Rv1, and NCI-H660) cell lines were cultured with 2% charcoal-stripped serum (CSS) instead of FBS. Similarly, BPH-1 and LNCaP cells were cultured with 2% FBS. For RWPE-1 and VCaP cells, drugs were screened in the same media as their maintenance growth conditions.

## Protocol application and expected results

This protocol was designed to provide an optimized and standardized approach to screen compounds across a panel of cell lines representing non-tumorigenic models and the spectrum of heterogeneous prostate cancer. To increase replicability and reproduciblilty within low/medium-throughput cell viability experiments, studies recommend cell line-specific optimization of variables such as drug concentration, media supplements, and cell seeding density for any specific viability assay [9, 16, 19]. As our system integrated both live-cell imaging and endpoint viability, a critical step of our protocol was to specifically optimize for sequential use of both approaches. The general requisites for optimization were that endpoint cell density must remain within the detectable linear range of the CellTiter-Glo assay, and that the IncuCyte ZOOM could verify continued proliferation of the control cells without achieving over-confluent within 72-hrs. Protocol Arm 1: "96-well plate cell growth optimization for integrated live-cell and endpoint viability drug screening assay" was designed so that in a single experiment (on a per cell line basis) the optimal media serum conditions, cell seeding density, and cell line compatibility for both platforms could be identified (Fig 1).

After establishing optimized cell growth conditions for each individual cell line, protocol Arm 2: "96-well plate OT-2 liquid handler integrated live-cell and endpoint viability drug activity screen" was designed to incorporate automation via the OT-2 liquid handler for cell seeding, five 1:3 serial drug dilutions, and drug delivery to cells before beginning real-time imaging via the IncuCyte ZOOM live cell analyzer and the CellTiter-Glo endpoint viability

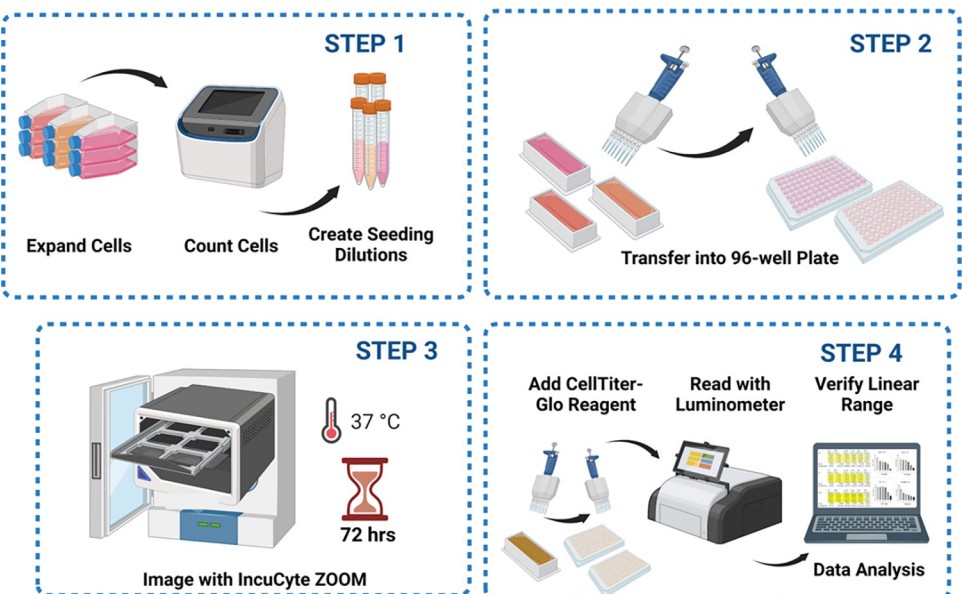

**Fig 1. Cell seeding optimization protocol schematic.** This figure was generated using the software platform Biorender.com.

assay to assess drug efficacy (Fig 2). Below we provide examples of how each arm of this protocol was conducted and expected data outputs.

## Arm 1: 96-well plate cell growth optimization for integrated live-cell and endpoint viability drug screening assay

Here we demonstrate execution of the protocol Arm 1: "96-well plate cell growth optimization for integrated live-cell and endpoint viability drug screening assay" using BPH-1 cells as a representative cell line. Before optimization BPH-1 cells seeded at 1000 cells/well in RPMI 1640 supplemented with 1% L-glutamine, 1% penicillin/streptomycin, and 10% fetal bovine serum (FBS) were ~90% confluent at the 72-hr experimental endpoint (Fig 3A). This level of confluence did not permit distinguishing between cells in densely populated areas or accurate assessment of confluence.A limitation of live-cell imaging to evaluate cell growth and drug efficacy is that cell confluence/density estimations become less accurate with higher confluence. This is due to the propensity of cells to clump (i.e. grow into the vertical plane as they proliferate). Furthermore, at higher endpoint confluence, cells treated with effective compounds tend to detach and this can obstruct visualization of adherent cells. These issues make it difficult to accurately capture representative images of attached cells, assess confluence, and measure drug efficacy.

To begin the optimzation procedure, a range of four seeding densities 250, 500, 750, and 1000 cells/well were compared in media supplemented with 2% or 5% FBS. Evaluation of live-cell imaging data showed that at 72-hrs cell confluence remained below the ~90% confluecnce observed in pre-optimized conditions in all cases. Additionally, when seeded below 1000 cells/well there were no differences in BPH-1 cells cultured in media supplemented with 2% or 5% FBS (Fig 3B and 3C). Although endpoint confluence when seeded at 750 and 1000 cells/well was lower than cells grown in media containing 10% FBS (Fig 3A), imaged sections displayed areas of clustered cell growth which risks loss of accuracy with IncuCyte ZOOM confluence tracking (Fig 3D).

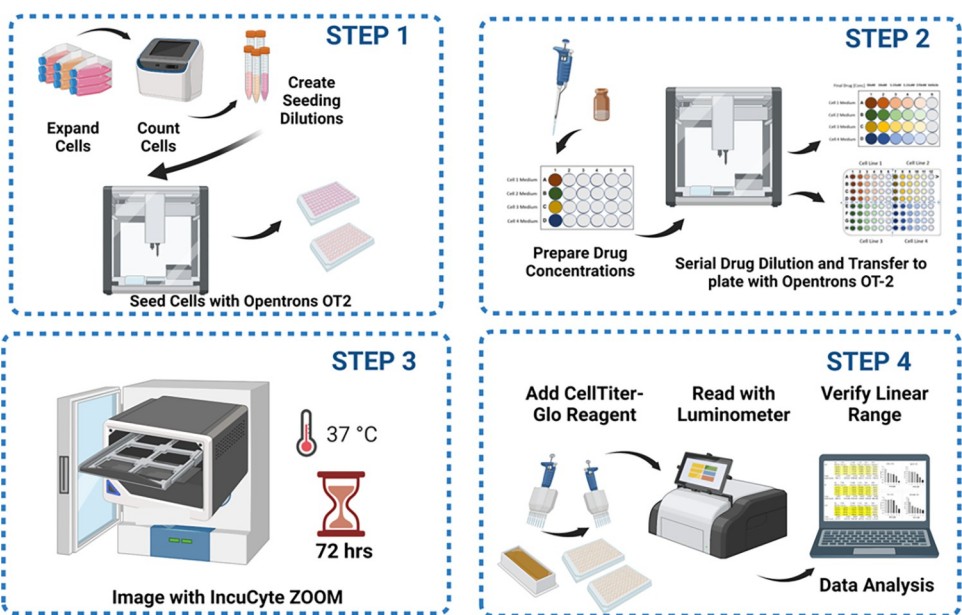

**Fig 2. Drug screening protocol schematic.** This figure was generated using the software platform Biorender.com.

At the 72-hr time point, cells were analyzed with the ATP viability CellTiter-Glo assay. Since cells were seeded in equal increments, we expected a linear relationship between ATP synthesis (raw luminescence values) and cell seeding densities. Thus, the first requirement of optimization was the determination of seeding densities and culture conditions that allowed for cell confluence after 72 hrs to be within the detectable linear range of the CellTiter-Glo ATP viability assay. Additionally, we used analyses of the live-cell imaging data to determine the cell-specific growth dynamics to ensure that cells did not exceed ~70% confluence. Another requirement was that the vehicle treated cells continued to proliferate through the duration of the experiment. In this example, considering IncuCyte confluence and adherence to detectable linear range for endpoint CellTiter-Glo viability assays, BPH-1 cells cultured in media supplemented with 2% FBS and seeded at 500 cells/well represented the optimal assay conditions. Optimization of all the cell lines within the tested panel followed the same rigorous procedure and only differed with cell-specific media and initial seeding densities. After optimization, growth curves of all cell lines were within the detectable linear range of the CellTiter-Glo assay (Fig 4) and at a similar growth rate and confluence at the experimental endpoint (Fig 5). Optimized conditions for 72-hr drug screening experiments in 96-well plates for nine human prostate cell lines are listed in Table 1.

### Arm 2: 96-well plate OT-2 liquid handler integrated live-cell and endpoint viability drug activity screen

Optimal seeding and growth conditions (Table 1) were applied to the Protocol Arm 2: "96-well plate OT-2 liquid handler integrated live-cell and endpoint viability drug activity screen". In this protocol, 8 cell lines were seeded into two 96-well plates (four cell lines per plate) via the Opentrons OT-2 liquid handler at the assay-specific seeding densities defined in the optimization protocol. The liquid handler robot equally dispensed 200ul of the cell suspension into the appropriate wells of an opaque white column, clear bottom 96-well plate (each cell line is seeded into one quadrant of one of the two 96-well plates). *Note: this type of plate

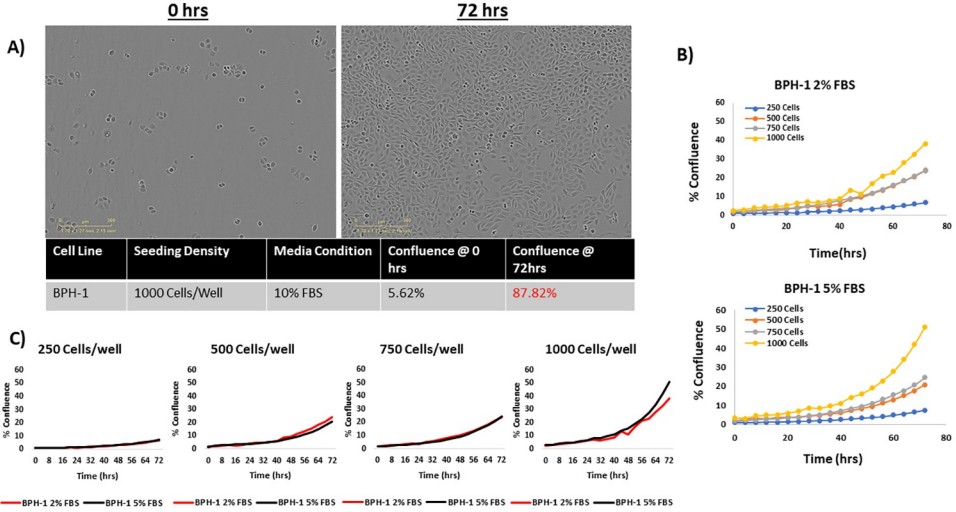

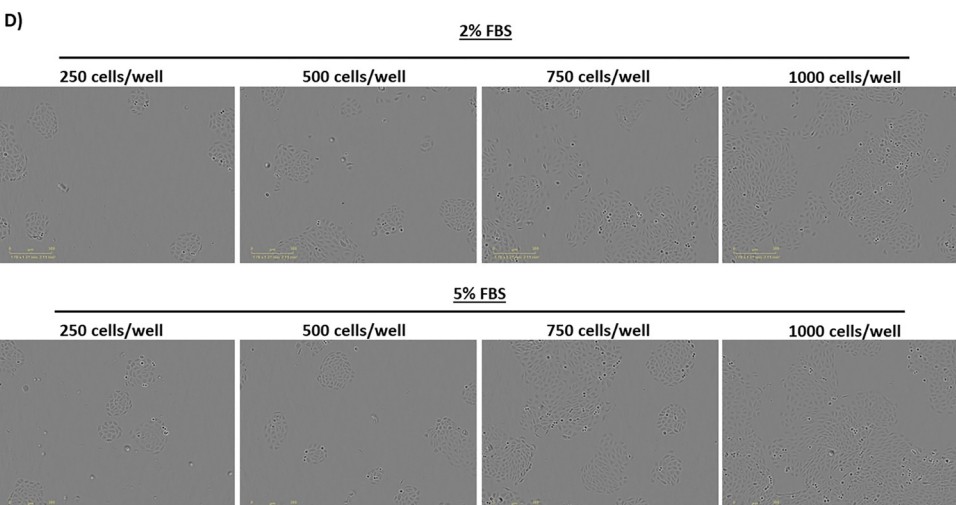

**Fig 3. Optimization of seeding density for non-tumorigenic BPH-1 cells.** Non-tumorigenic human prostate BPH-1 cells were cultured in full serum (media supplemented with 10% FBS) and seeded at 1000 cells/well (replicates of 4 wells imaged 4 times per 4 hrs) in a 96-well plate. Real-time imaging tracked via the IncuCyte ZOOM live cell analyzer documented and evaluated **A)** cell vitality, morphology, and confluence over a 72-hr timespan. Separately, BPH-1 cells were seeded at 250 cells/well, 500 cells/well, 750 cells/well, 1000 cells/well at both 2% FBS (replicates of 12 wells imaged 4 times per 4 hrs) and 5% FBS (replicates of 12 wells imaged 4 times per 4 hrs) and cell confluence was tracked quantifiably as an average of each replicate **B-C)** and **D)** visually through 72-hrs live-cell imaging via IncuCyte ZOOM.

is necessary as the IncuCyte ZOOM images from the bottom of the plate, and the CellTiter-Glo assay requires white opaque wells to confine bioluminescence. After the cells settled overnight, drugs were manually prepared at the desired highest concentration (30μM in our system) and transferred to the liquid handler with the cell plates for 1:3 serial dilutions and drug delivery. The concentration range was 0 (DMSO Vehicle), 370nM, 1.11μM, 3.33μM, 10μM, and 30μM. The original "seeding" media were removed by the robot and replaced with 100ul (in quadruplicate replicates per concentration per cell line) of the appropriate drug concentration. During the development of the Opentrons OT2 protocol, we determined that removing/adding media to cells at a flow rate of 5 μL/sec with the pipette tip positioned ~1mm off center prevented significant cell loss/detachment. After drug transfer, plates were transported into

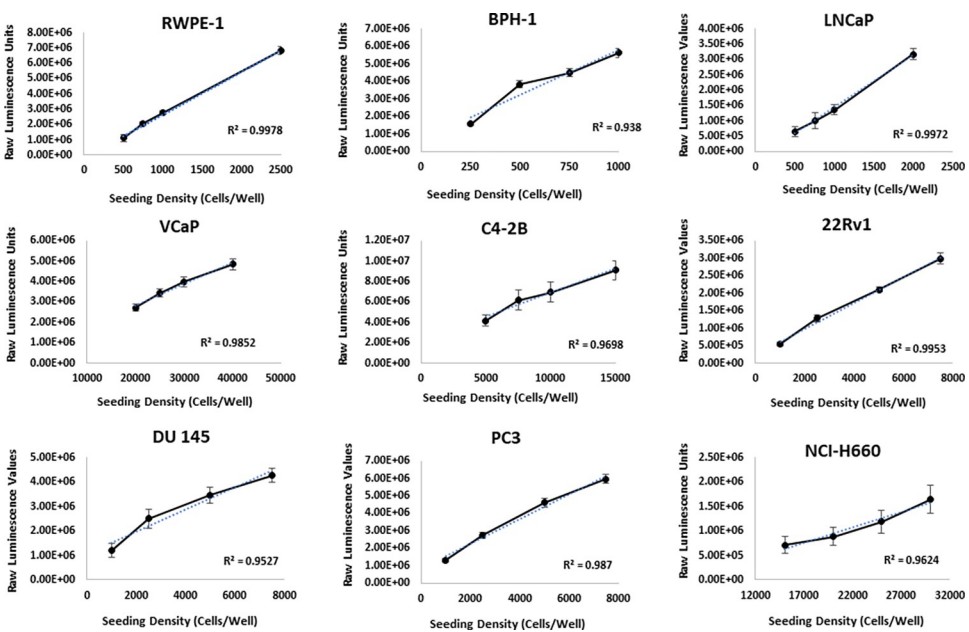

**Fig 4. Cell seeding density optimization for endpoint viability assay linear range.** A panel of human prostate cells was seeded at a range of cell-specific densities in 96-well plates in which all CRPC (C4-2B, PC3, DU 145, 22Rv1, and NCI-H660) cell lines were cultured with 2% charcoal-stripped serum (CSS), BPH-1 and LNCaP cells were cultured with 2% FBS, RWPE-1 cells were cultured in standard supplemented KSFM growth media, and VCaP cells with 10% FBS. Cells were analyzed at 72-hrs via cell-titer glo viability assay. Raw luminescence values were plotted against seeding density and linearity ($R^2 \geq 0.9$) was established through linear regression.

the IncuCyte ZOOM to begin 72-hr live-cell tracking. Any changes in cell growth rates, confluence, morphology, or cell death were archived throughout drug treatment. At the 72-hr endpoint, the bottom of each plate was covered with opaque white tape (as recommended by

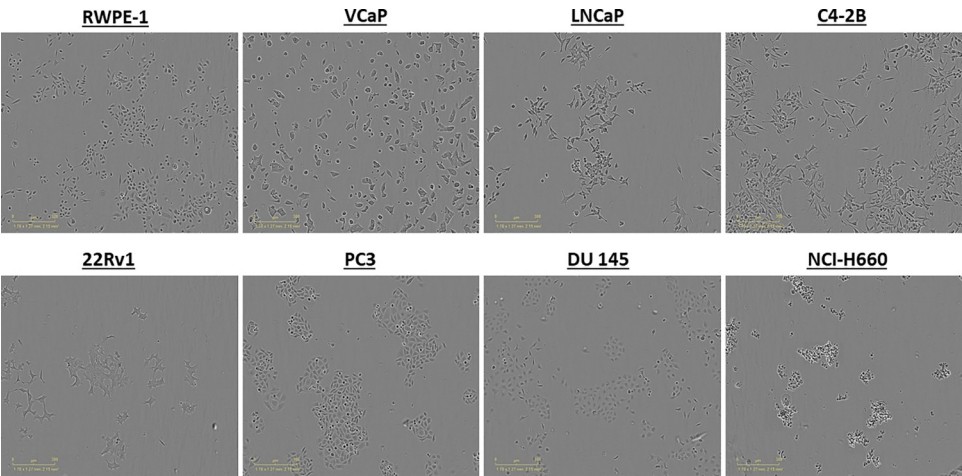

**Fig 5. Representative experiment: Cell growth comparable across human prostate cell panel after optimization.** A panel of human prostate cells was seeded at optimized conditions (described in Table 1) via the OT-2 liquid handler and the next day diluted DMSO (to 0.06%) with cell-specific media containing 2% CSS (all CRPC), 2% FBS (LNCaP), 10% FBS (VCaP) and RWPE-1 (standard supplemented KSFM media). Images were tracked via IncuCyte ZOOM for 72-hrs. Representative images for each cell line are displayed.

**Table 1. Optimized 96-well plate conditions for human prostate cell drug screening protocol.**

| Cell Line | Cell Type | Seeding Density (Cells/Well) | Media Type | Media Condition | Patient Source |
|-----------|-----------|------------------------------|------------|-----------------|----------------|
| RWPE-1 | Non-Tumorigenic | 2,500 | KSFM | Supplemented | Primary Prostate |
| BPH-1 | Non-Tumorigenic | 500 | RPMI | 2% FBS | Primary Prostate |
| VCaP | ADPC AR+ ARv7+ | 50,000 | DMEM + Glutamax | 10% FBS | Lumbar Vertebra Metastasis |
| LNCaP | ADPC AR+ | 1,000 | RPMI | 2% FBS | Lymph Node Metastasis |
| C4-2B | CRPC AR+ | 5,000 | DMEM | 2% CSS | Derived from LNCaP cell line |
| 22Rv1 | CRPC AR+ ARv7+ | 2,500 | RPMI | 2% CSS | Derived from CWR22 xenograft (Bone Metastasis) |
| DU 145 | CRPC AR Null | 2,500 | RPMI | 2% CSS | Brain Metastasis |
| PC3 | CRPC AR Null | 2,500 | RPMI | 2% CSS | Bone Metastasis |
| NCI-H660 | NEPC AR Null | 30,000 | RPMI-HG | 2% CSS | Lymph Node Metastasis |

Promega) to prevent loss of detectable luminescence or contamination between wells. Then cell viability was quantified via the CellTiter-Glo assay.

## Data acquisition and analysis

By integrating live-cell imaging and endpoint viability assessments, the data generated through this optimized protocol provided multiple informative/comparative metrics with which to assess drug efficacy. In addition to enhanced quality control, live-cell imaging displayed any morphological changes that cells may have undergone because of treatment, and gave qualitative insight to cell death, morphological changes, or cytostatic vs cytotoxic effects throughout the 72-hr drug treatment. As an example, we demonstrate application of the drug screening protocol with C4-2B CRPC cells treated with first-in-class BET and CBP/p300 dual inhibitor EP-31670. EP-31670 has proven effectiveness in prostate cancer [20, 21] and is currently undergoing phase 1 clinical trials for treatment of patients with CRPC (Clinical Trial ID: NCT05488548).

In this example, we used the C4-2B cell line to represent endpoint imaging, although these data were gathered for each cell line of the human prostate cell panel during each experimental run. The IncuCyte ZOOM also tracked changes in confluence throughout the experiment and allowed for evaluation of the morphological effects of tested compounds on cells. Real-time imaging demonstrated EP-31670 efficacy in prostate cancer as CRPC C4-2B cell proliferation was reduced even in the sub-micromolar range (Fig 6A). In addition to representing drug efficacy as a direct function of changes in cell growth (compared to vehicle) across a drug concentration range, our approach also employs growth rate inhibition (GR) to produce drug response metrics. GR curves uniquely utilize cell growth dynamics to represent drug response. One method to calculate the values used to generate GR curves requires a time-dependent component such as the growth rate of cells in experimental conditions and a quantification of cellular response to tested drugs (i.e., endpoint confluence or cell metabolism). Our protocol used the IncuCyte ZOOM live-cell analyzer to determine vehicle treated cell line specific growth rates through 72-hrs in optimized conditions. We then used the measures of cell metabolism across the tested concentration range (vehicle-30uM) from the CellTiter-Glo endpoint viability assay to represent cell-specific drug response. For each experiment, the growth rates and drug responses for each cell line were assessed and integrated together to serve as the input values for the online GR calculator Tool provided by the Sorger Lab at Harvard Medical School [22] [http://www.grcalculator.org/grcalculator/]. In most cases drug response as assessed by endpoint confluence produced similar results to CellTiter-Glo (S1 Fig) and sufficed as input data for GR curves. However, by assessing drug response through ATP-based cell

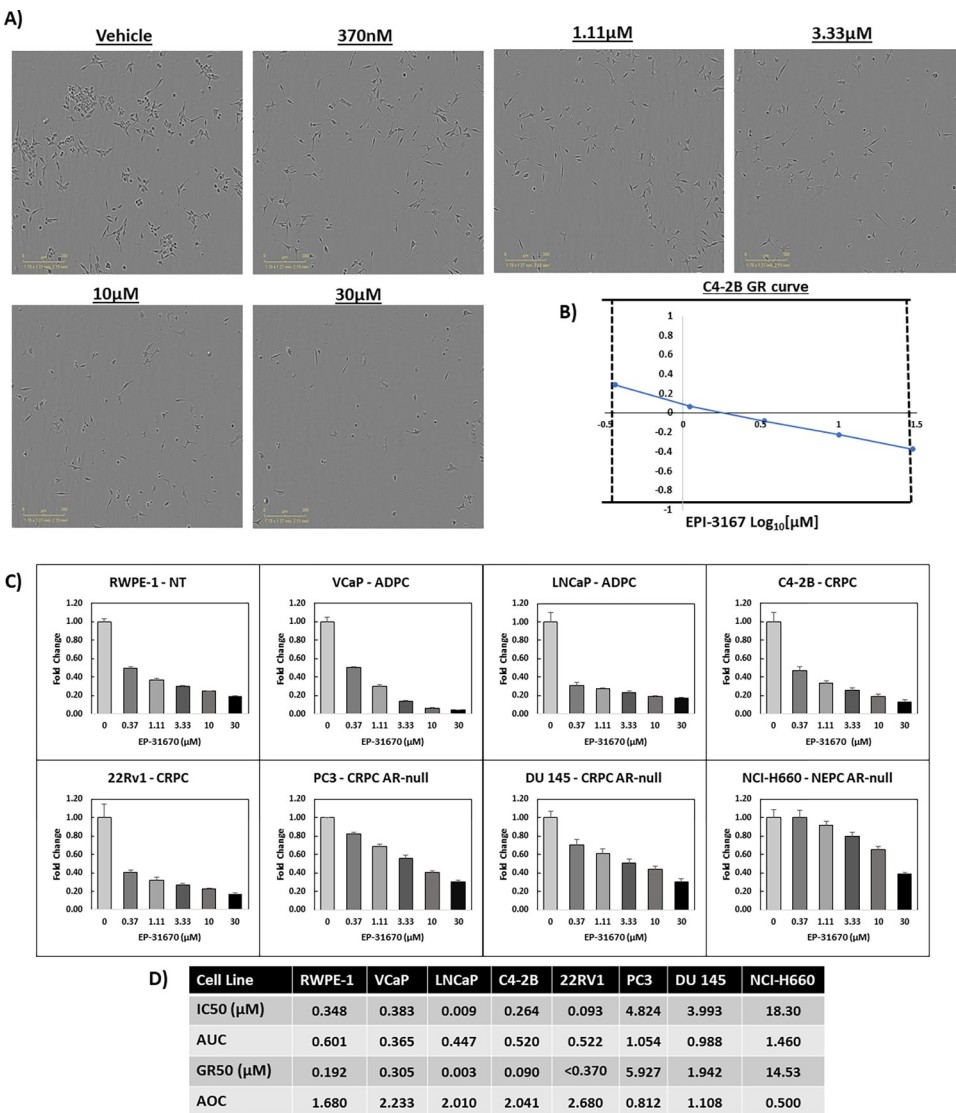

**Fig 6. Representative human prostate cell panel dose-response to bromodomain inhibitor EP-31670.** A panel of human prostate and cancer cells were seeded in a 96-well plate at optimized conditions (described in Table 1) via the OT-2 liquid handler. After the cells adhered to the 96-well plate overnight, the first-in-class BET and CBP/p300 dual inhibitor EP-31670 was diluted via the OT-2 liquid handler to concentrations of 0 (DMSO vehicle), 370nM, 1.11μM, 3.33 μM, 10 μM, and 30 μM (replicates of four wells per concentration) and cells were treated. The IncuCyte ZOOM live cell analyzer tracked cell growth (4 images per 4 hrs per replicate) over a 72-hr time span then an endpoint CellTiter-Glo cell viability assay was completed. **A)** Representative images of the proliferative response of CRPC C4-2B cells to EP-31670 at 72-hrs is displayed and **B)** the C4-2B drug response based on the endpoint viability assay and calculated cell-specific growth rates (via live-cell imaging) were integrated to generate a growth rate inhibition (GR) curve at 72-hrs. **C)** Dose-response curves for non-tumorigenic (NT) prostate cells RWPE-1, androgen-dependent prostate cancer (ADPC) LNCaP and VCaP, castration-resistant prostate cancer (CRPC) C4-2B and 22Rv1,AR-null CRPC DU 145 and PC3, and neuroendocrine prostate cancer (NEPC) NCI-H660 based on CellTiter-Glo endpoint viability at 72-hrs. Fold change growth normalized to the lowest drug concentration 0 (vehicle) for each concentration was calculated ± SD (quadruplicate replicates). **D)** A Summary table including calculated IC50, area under the curve (AUC), GR50, and area over the curve (AOC) values for tested human prostate cell lines was generated using values from the online GR calculator tool.

viability we avoid possible false negative classifications which can occur when imaging platforms erroneously include dead/floating cells in confluence assessment (S2 Fig).

GR curves have advantages over standard dose-response curves. Standard sigmoidal dose-response curves are derived by comparing cell counts (typically via hemocytometry or endpoint viability analyses) of drug treated cells to their untreated controls. These curves are still the industry standard and analyses of them generate evaluative metrics such as IC50 (the concentration of drug that results in 50% of the vehicle control cell number) and AUC (area under the curve). Internally conducted experiments reliant on the traditional sigmoidal curves can provide robust data outputs. However, many studies which rely solely on these metrics are plagued by lack of reproducibility across different labs presumably due to variations in seeding densities, media conditions, and cell doubling times [23]. By comparing cell growth rates in assay-specific conditions for treated and untreated cells, GR curves can be generated which control for many sources of variability and provides drug response evaluative metrics adaptable for reproducible drug screening efforts. [22] In this example, GR values (ranging from -1 to 1) were assigned to each tested concentration, ultimately creating a curve where concentrations with GR $\leq$ 0 indicate cytotoxic effects, GR = 0 for cytostatic, and GR > 1 reduced proliferation (Fig 6B).

We screened EP-31670 in our panel of representative human prostate cells models: RWPE-1 (non-tumorigenic), LNCaP and VCaP (ADPC), C4-2B and 22Rv1 (CRPC), DU 145 and PC3 (androgen receptor-null CRPC), and NCI-H660 (NEPC). Our screen demonstrated that this compound had efficacy in both ADPC and CRPC (Fig 6C, S4 Fig) which supports data demonstrating the interplay of AR and bromodomain containing proteins in prostate cancer [24–26], but also confirmed findings that highly resistant AR-null cell lines (PC3, DU 145 and NCI-H660) are also sensitive [21] albeit to a lesser extent. For each experiment, we summarized the data as representative dose-response curves (generated from the CellTiter-Glo data) and tables (generated with integrated CellTiter-Glo and IncuCyte data) with the calculable and comparable metrics IC50, area under the curve (AUC), GR50, and area over the curve (AOC) for evaluation and comparison of drug performance (Fig 6D).

The metrics AUC (area under the curve) and AOC (area over the curve) are calculations of the area beneath the traditional sigmoidal dose response curves or the area above the GR dose response curve. We incorporated AUC and AOC values because although IC50 and GR50 are the standard for drug efficacy data, they have a critical limitation when used in dose-limited comparative screening efforts. If the tested drugs do not inhibit cell growth sufficiently within the tested drug concentration range, then these values cannot be calculated. AUC and AOC values are derived from the integral of the sigmoidal dose response curve (AUC) or GR curve (AOC) from the lowest tested concentration (0 in our screen) concentration to the max concentration (30μM in our screen). In the case of AUC, a completely ineffective compound at every concentration will have a linear curve and maintain a fold change value of 1 and have a higher AUC. Conversely, a compound that is effective at the lowest tested concentrations will have a lower AUC. The closer to 0 the AUC the more effective the compound is. When evaluating GR curves, in which x-axes range from -1 through 1, the most effective compounds will have values closer to -1 at cytotoxic concentrations and ineffective compounds will be closer to 1. Since GR curves calculate area over the curve, the most effective compounds will have a larger AOC value. When evaluating compound efficacy in cases where IC50 or GR50 cannot be calculated, comparing the AUC and AOC values permits quantifiable assessments across cell lines.

## Validation of approach

For validation, we have included data from a screen of insulin-like growth factor-1 receptor (IGF-1R) small molecule inhibitor linsitinib as an example. IGF-1R levels increase as prostate

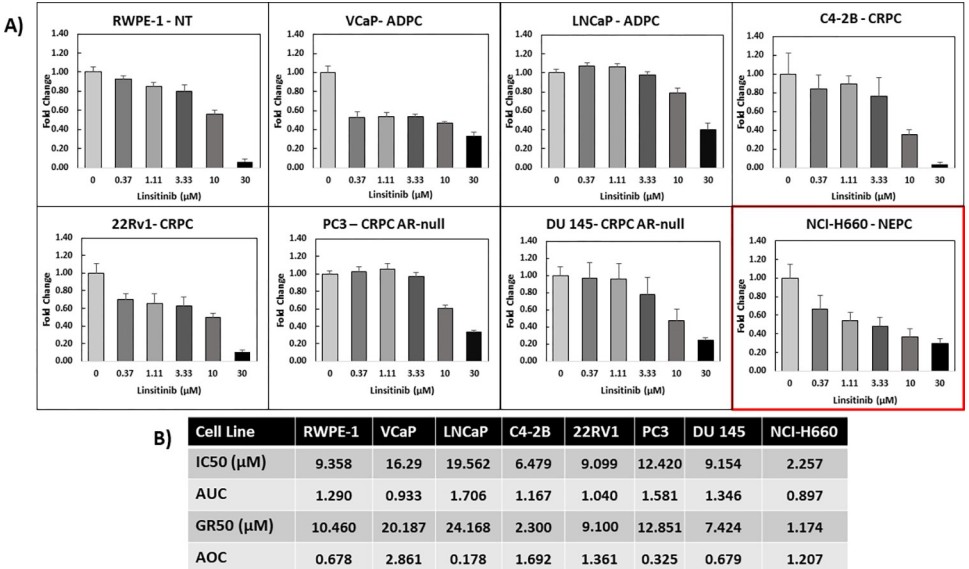

**Fig 7. Validation of effective compounds in advanced prostate cancer: IGF-1R inhibitor.** Non-tumorigenic (NT) prostate epithelial cells RWPE-1, androgen-dependent prostate cancer (ADPC) LNCaP and VCaP, castration-resistant prostate cancer (CRPC) C4-2B and 22Rv1, AR-null CRPC DU 145 and PC3, and neuroendocrine prostate cancer (NEPC) NCI-H660 cells were seeded via the OT-2 liquid handler at optimized conditions (described in Table 1). The cells adhered to the 96-well plate overnight, cells were screened with IGF-1R inhibitor (linsitinib) via optimized drug screen protocol at 6 concentrations: 0 (DMSO vehicle), 370nM, 1.11μM, 3.33 μM, 10 μM, and 30 μM (quadruplicate replicates per concentration). **A)** Dose -response curves were generated based on CellTter-Glo endpoint viability at 72-hrs. Fold change growth normalized to the vehicle concentration was calculated ± SD. **B)** Complete analysis was summarized into a table after integrating cell growth rates and endpoint viability-based drug response via the Online GR Calculator tool calculated IC50, area under the curve (AUC), GR50, and area over the curve (AOC) values for tested prostate cancer cell lines.

cancer progresses from ADPC to CRPC and studies have shown promising anti-proliferative effects of IGF-1R inhibitors in both in vitro and in-vivo settings [27–31]. Our screen confirmed that linsitinib was effective in ADPC (particularly VCaP) and CRPC (Fig 7A) and additionally uncovered that the growth of highly incurable NEPC-type cell line NCI-H660 cells was reduced by IGF-1R inhibition even in the lower micromolar and nanomolar concentration ranges. Interestingly, NCI-H660 cells were more sensitive to linsitinib than other tested non-tumorigenic prostate, ADPC, and CRPC cell lines. (Fig 7B, S4 Fig). We also demonstrate differential drug efficacy for the CYP17 inhibitor / AR antagonist abiraterone acetate but only at the 10μM and 30μM concentrations for the AR-expressing non-tumorigenic cell line (RWPE-1) and the AR-expressing (especially LNCaP and VCaP) compared to the AR-null PC cell lines (PC3, DU 145, and NCI-H660) (S3 Fig). AR signaling inhibitors (ARSIs) including abiraterone acetate are typically used in the ≥ 10μM range and cause mostly cytostatic effects (cell death is less pervasive) [32, 33]. Furthermore, ARSIs typically require longer times (more than 72 hrs) to see effects on cell growth [34]. For these reasons, ARSIs as single compounds do not produce robust cell growth inhibitory effects with this platform.

Incorporating real-time imaging to enhance pharmacological screens also identified cases where although confluence and endpoint viability assays were aligned and suggested no effect by the tested drug, there were clear morphological changes which prompt a deeper investigation into how the tested drug affects cells. In the provided example, we tested paprotrain, an inhibitor of the kinesin KIF20A which has recently been identified as a key member of a prognostic prostate cancer oncogenic signature and a driver of prostate cancer progression [35, 36]. In AR-null CRPC DU 145 cell lines, both endpoint confluence and viability showed a

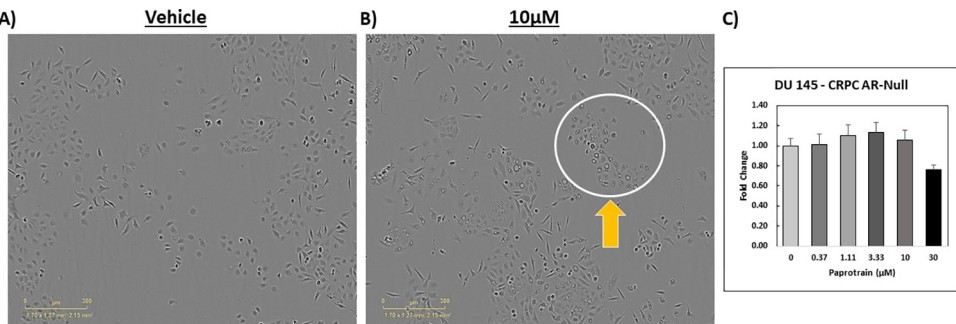

**Fig 8. Integrated approach identified morphological differences undetectable in viability assay.** DU 145 cells were seeded at 2,500 cells/ well via the OT-2 liquid handler and became adherent to the 96-well plate overnight. Then cells were treated (via the OT-2) with KIF20A small-molecule inhibitor paprotrain at a concentration range of 0μM -30μM. Images were tracked for 72-hrs then an endpoint cell viability assay was conducted. Comparing representative images of DU 145 treated at **A)** 0μM (DMSO vehicle) and **B)** 10μM paprotrain at 72-hrs revealed unique morphological differences (See white circle and yellow arrow) that were not reflected in confluence or **C)** in the CellTiter-Glo endpoint viability dose-response curve (fold change growth normalized to the lowest drug concentration 0 (vehicle) for each concentration was calculated ± SD.

minimal increase in proliferation and cell metabolism (Fig 8C). Upon evaluation of the live-cell images, morphological differences between vehicle and cells treated with paprotrain most profoundly at 10μM (Fig 8A and 8B) were apparent. Our integrated screen mitigated this type of classification of compounds as false negatives as can occur with single assay approaches.

## Limitations

The major limitation of the "96-well plate OT-2 liquid handler integrated live-cell and endpoint viability drug activity screen" protocol is that it takes ~65 minutes per 96-well plate to complete the Opentrons OT-2 liquid handler drug dilution and drug treatment steps. Currently, the drug dilution step takes ~40 minutes to complete leaving ~25 minutes for the drug transfer to cells step. There are two rate limiting actions in this section of the protocol. The first is that for the drug dilution steps, additional mixing steps had to be added to ensure uniform concentrations. This problem can be circumvented in the future by using a 1mL pipette channel for the OT-2 liquid handler. The second challenge was removing all the "seeding media" from each well of the 96-well plate without disturbing attached cells which required the pipette speed to be significantly lowered which prolonged the process. In the future, use of a 1mL pipette channel to approximately halve the duration of the drug dilution steps will be tested.

An additional limitation to using this protocol is the inability to automate organizing pipette tips into the specific pattern required for seeding cells into their specified quadrants. To test four cell lines per 96-well plate at six different concentrations, cell lines must be seeded separately within a specified quadrant of the plate. The liquid handler system utilizes an 8-channel pipettor attachment which is designed to pick up pipette tips for all 8-channels. The Opentrons OT-2 has a limited range of X-Y planar motions making it impossible to create a script that permits automated seeding into four wells at a time while using the 8-well 300μl pipette channel compatible with the OT-2. For this reason, before seeding users must rearrange tip rack patterns in a sterile environment which consumes time and increases likelihood of user error.

## Discussion/Conclusions

We developed and validated a protocol which identified optimal conditions with which to conduct low to medium-throughput integrated live-cell imaging and endpoint viability drug

screens. While other studies have demonstrated the benefit of combining both live-cell imaging and endpoint viability assays [15, 18], our adaptable protocol first uniquely establishes cell seeding densities and media conditions optimized specifically for compatibility with sequential integrated real-time and endpoint cell viability for drug screens and then executes comparative drug screens across a panel of diverse cell lines. In the protocol, we utilized the OpenTrons OT-2 and IncuCyte ZOOM live-cell analyzer platforms however, the protocol was created with enough versatility and detailed instructions so that users who have other platforms and study other cell models may readily adjust it to fit their needs. For each experimental run using this protocol, three key data types were obtained: 1) cell-specific drug response represented as endpoint cell viability and measured by the CellTiter-Glo assay. 2) Live-cell imaging via the IncuCyte ZOOM platform which quantifiably tracks real-time confluence, cell growth dynamics, and permits qualitative assessments of morphological changes as a response to drug treatment. 3) A suite of data including drug efficacy metrics IC50, AUC, GR50, and AOC which were calculated through the integration of both cellular growth rates as determined through the IncuCyte ZOOM imaging platform and the endpoint cell metabolism ATP viability assay data. These drug screens provide all necessary data to draw conclusions about drug efficacy across a heterogeneous panel of cell lines while avoiding the shortcomings of each individual approach including potential false negative classification as discussed above.

In this study, we validated that first-in-class BET and CBP/p300 dual inhibitor EP-31670 reduced ADPC and CRPC cell growth. Previous studies have showed that EP-31670 was effective in CRPC including AR-null CRPC cell lines PC3 and DU 145 [20, 21]. Interestingly, our comprehensive screen demonstrated that AR-null CRPC cell lines were affected to a lesser extent than AR-expressing CRPC or ADPC cell lines tested. This may be due to a greater reliance on interplay between AR and bromodomain proteins making these cells more vulnerable to BET inhibition. Additionally, all tested AR positive cell lines, including non-tumorigenic RWPE-1 cells were sensitive to EP-31670 more than likely due to the documented interplay of AR signaling and bromodomain proteins [24–26]. We also observed that at the lowest tested EP-3160 concentration (370nM), RWPE-1 cells were slightly less sensitive than ADPC LNCaP, CRPC C4-2B, and AR-V7 driven 22Rv1 cell lines. Further testing over a broader nanomolar range of EP-31670 may reveal a therapeutic window.

Furthermore, we found that IGF-1R inhibitor linsitinib was a potential pharmacological agent against highly drug-resistant (supported by our unpublished results from another study) NEPC NCI-H660 cells. Other work has suggested that NEPC might be sensitive to IGF-1R inhibition [37, 38], but this study is the first to demonstrate proliferative inhibition of the highly resistant human NEPC NCI-H660 cells by an IGF-1R inhibitor. Our screen identified that NCI-H660 cells were more sensitive to linsitinib than other tested non-tumorigenic, ADPC, and CRPC cell lines. These data prompt additional drug screens and investigation into the roles and mechanisms of IGF-1R in NEPC. Lastly, we showed how this integrated screen can identify cell morphological changes when there is no evident difference in cell viability or growth mitigating potential false negative conclusions.

To date, over 500 drug/cell combinations have been tested using this approach producing thousands of evaluative drug efficacy data points and demonstrating feasibility of medium-throughput analyses by an individual lab (manuscript in preparation). Throughout these studies, we identified compounds which were "selective responders" (defined by at least one of nine tested cell lines having an IC50 or GR50 value below 30μM), "all-responders" (defined by all tested cell lines having an IC50 or GR50 value below 30μM), and "non-responders" (defined by no tested cell lines having IC50 or GR50 value below 30μM). We designed and optimized this platform to screen single compounds with a future goal of expanding this type of analysis to drug combinations.

In summary, this adaptable protocol overcomes limitations in traditional comprehensive drug screens which use either real-time cell viability or endpoint cell viability assays to evaluate drug response. While live-cell imaging or endpoint viability assays alone can often identify cellular responses to drugs, we show here the benefits of a platform that integrates both approaches for use in multiple cell lines. Adoption of automated robotics increases precision, reproducibility, and reduces untraceable human error. This platform identifies drug screening "hits" through tracking cell growth and confluence throughout each 72-hr drug treatment prior to the cell metabolism-based endpoint viability assays. This integrated approach expands the range of output data to include real-time representation of morphological changes, cell confluence/growth rates, and quantifiable pharmacological performance metrics including (but not limited to) traditional dose-response curves, growth rate inhibition (GR) dose-response curves, IC50, and GR50. This protocol produces data and analytical metrics that may be readily adapted for use by individual laboratories to streamline consistently, robustly, and reproducibly low to medium-throughput drug screening efforts.

## Supporting information

**S1 File. Automation, live-cell imaging, and endpoint cell viability for 96-well plate prostate cancer drug screens.** Step-by-step protocol, also available on protocols.io: dx.doi.org/10.17504/protocols.io.x54v9dojzg3e/v1.
(PDF)

**S2 File.**
(PDF)

**S1 Fig. Demonstrating consistency between calculated drug response when comparing endpoint confluence to endpoint ATP viability.** This figure was generated using data from the same experiment described in Fig 6. Non-tumorigenic (NT) prostate cells RWPE-1, androgen-dependent prostate cancer (ADPC) LNCaP, castration-resistant prostate cancer (CRPC) C4-2B, and AR-null CRPC PC3 cells were seeded in a 96-well plate at optimized conditions and treated for 72-hours (as described in Fig 6). These representative curves for each cell line were generated to demonstrate consistency in assessing drug response via endpoint confluence with the IncuCyte ZOOM and endpoint viability using CellTiter-Glo ATP viability data at 72-hrs for the tested cell lines. Fold change growth normalized to the lowest drug concentration 0 (vehicle) for each concentration was calculated ± SD (quadruplicate replicates).
(TIF)

**S2 Fig. Visualization of IncuCyte ZOOM false negative classification.** This figure was generated using data from the same experiment described in Fig 6. ADPC VCaP cells were seeded in a 96-well plate at optimized conditions and treated for 72-hours (as described in Fig 6). VCaP cell response to EP-31670 was estimated via **A)** endpoint confluence with the IncuCyte ZOOM and endpoint viability using CellTiter-Glo ATP viability data at 72-hrs for the tested cell lines. **B)** Comparing representative images from VCaP cells treated with vehicle vs 30μM show that at higher concentrations live-cell imaging can erroneously interpret dying/floating cells as viable cells (yellow outline).
(TIF)

**S3 Fig. Drug screen of AR antagonist abiraterone acetate.** A panel of human prostate and cancer cells were seeded in a 96-well plate at optimized conditions (described in table 1) via the OT-2 liquid handler. After the cells adhered to the 96-well plate overnight, the AR antagonist abiraterone acetate was diluted via the OT-2 liquid handler to 0 (DMSO vehicle), 370nM,

1.11μM, 3.33 μM, 10 μM, and 30 μM (replicates of four wells per concentration) and cells were treated. Endpoint CellTiter-glo cell viability assay was completed. Dose-response curves for non-tumorigenic (NT) prostate cells RWPE-1, androgen-dependent prostate cancer (ADPC), LNCaP and VCaP, castration-resistant prostate cancer (CRPC) C4-2B and 22Rv1, androgen receptor (AR-null CRPC) DU 145 and PC3, and neuroendocrine prostate cancer (NEPC) NCI-H660 based on CellTiter-glo endpoint viability for at 72-hrs. Fold change growth normalized to the lowest drug concentration 0 (vehicle) for each concentration was calculated. ± SD (quadruplicate replicates).
(TIF)

**S4 Fig. Representation of data from Figs 6 and 7 as estimated dose response curves.** A panel of human prostate and cancer cells were seeded at optimized conditions and treated for 72-hours (as described in Figs 6 and 7) with **A)** the first-in-class BET and CBP/p300 dual inhibitor EP-31670 or **B)** the IGF-1R inhibitor linsitinib, respectively. For visualization, dose response curves were estimated using CellTiter-Glo endpoint viability at 72-hrs. Cellular drug response data were fit into a non-linear sigmoidal regression curve via and plotted together. Since the protocol screens only 5 drug concentrations, an algorithm to estimate these curves assigned values of $Log(M) = -9$ and $Log(M) = -1$ respectively to represent the two extremes of the curves (completely ineffective concentration and completely effective concentration).
(TIF)

## Acknowledgments

The authors are grateful to Dr. Claes Wahlestedt (University of Miami, Center for Therapeutic Innovation) for providing EP-31670, Dr. Simon Hayward (Northshore University Health System, Cancer Biology) for BPH-1 cells, and Dr. Nahuel Peinetti for assistance with data representation and analysis.

## Author Contributions

**Conceptualization:** Rolando D. Z. Lyles, Maria J. Martinez, Stephan Schürer, Kerry L. Burnstein.

**Data curation:** Rolando D. Z. Lyles, Maria J. Martinez.

**Formal analysis:** Rolando D. Z. Lyles, Maria J. Martinez.

**Funding acquisition:** Stephan Schürer, Kerry L. Burnstein.

**Investigation:** Rolando D. Z. Lyles, Maria J. Martinez.

**Methodology:** Rolando D. Z. Lyles, Maria J. Martinez, Benjamin Sherman, Kerry L. Burnstein.

**Project administration:** Rolando D. Z. Lyles, Benjamin Sherman.

**Supervision:** Stephan Schürer, Kerry L. Burnstein.

**Validation:** Rolando D. Z. Lyles, Maria J. Martinez.

**Writing – original draft:** Rolando D. Z. Lyles.

**Writing – review & editing:** Rolando D. Z. Lyles, Maria J. Martinez, Benjamin Sherman, Stephan Schürer, Kerry L. Burnstein.

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
