## [Decision Letter · Decision Letter 0]

24 Apr 2023

PONE-D-23-06181Automation, live-cell imaging, and endpoint cell viability for prostate cancer drug screensPLOS ONE

Dear Dr. Burnstein,

Thank you for submitting your manuscript to PLOS ONE. After careful consideration, we feel that it has merit but does not fully meet PLOS ONE’s publication criteria as it currently stands. Therefore, we invite you to submit a revised version of the manuscript that addresses the points raised during the review process. In addition to providing requested clarifications, please discuss in more detail the advantages of your protocol compared to the high throughput screening techniques with a single treatment option. Perhaps  integrating imaging and ATP levels of a combination treatment would reveal an advantage, as suggested by 2 reviewers.

We look forward to receiving your revised manuscript.

Kind regards,

Irina U Agoulnik, Ph.D.

Academic Editor

PLOS ONE

Reviewers' comments:

Reviewer's Responses to Questions

**Comments to the Author**

1. Does the manuscript report a protocol which is of utility to the research community and adds value to the published literature?

Reviewer #1: Yes

Reviewer #2: Yes

Reviewer #3: Yes

Reviewer #4: Yes

2. Has the protocol been described in sufficient detail?

To answer this question, please click the link to protocols.io in the Materials and Methods section of the manuscript (if a link has been provided) or consult the step-by-step protocol in the Supporting Information files.

The step-by-step protocol should contain sufficient detail for another researcher to be able to reproduce all experiments and analyses.

Reviewer #1: Yes

Reviewer #2: Yes

Reviewer #3: Partly

Reviewer #4: Yes

3. Does the protocol describe a validated method?

Reviewer #1: Yes

Reviewer #2: Yes

Reviewer #3: Yes

Reviewer #4: Yes

4. If the manuscript contains new data, have the authors made this data fully available?

Reviewer #1: Yes

Reviewer #2: Yes

Reviewer #3: Yes

Reviewer #4: No

**5. Is the article presented in an intelligible fashion and written in standard English?**

Reviewer #1: Yes

Reviewer #2: Yes

Reviewer #3: Yes

Reviewer #4: Yes

6. Review Comments to the Author

Reviewer #1: The authors have established a semi-automated method to screen drugs for the treatment of prostate cancer (PCa). The method employs a combination of microscopic live cell imaging and endpoint proliferation assay. The authors used a BET and CBP/p300 inhibitor and an IGF-1R inhibitor to validate the method.

1. All data presented are from end point assay (cell titer glo), which is just a manual method and has nothing to do with semi automation. It is unclear how the end point assay in this method differs from traditional end point assay. No data interpretation or results from imaging have been provided in detail. May be a correlation between cell titer glo and imaging-based data analysis be helpful to determine the consistency between the methods.

2. AR antagonist screen will be helpful to determine if the method differentiates between AR-positive and -negative cells in terms of growth inhibitory effect.

Reviewer #2: The authors describe an adaptable strategy to optimize assay-specific cell growth conditions and simultaneously assess drug efficacy across multiple cell lines that can be employed across other cancer model systems. While there are numerous publications on drug screening methodologies, there remains room for studies that comprehensive present a detailed and informative approach, especially one that highlights advantages of certain approaches.

There are some issues with the current manuscript that are described below.

It is not clear how “optimal assay conditions” are defined. Is it the linear relationship between cell metabolism and seeding density, as shown and implied in F4? That ignores the dynamic range of the assay to measure growth effects for drug screening. The abstract claims to have demonstrated reliability and reproducibility in the reported screening approach, however, no measurements of reliability or repeatability are presented. These latter quantifiable metrics would be strong measures to define optimal conditions.

The cell numbers plated and growth conditions lead to low confluence at the end of 72hrs. The advantages of this over higher confluence at 72 hrs (the premise from F3A) is unclear.

Was there cell loss with removing media following 24hrs of plating. Experience of many screening labs indicates that removing media from 96 well plates can lead to loss of cells.

“Real-time imaging demonstrated EP-31670 efficacy in prostate cancer as CRPC C4-2B cell proliferation was reduced even in the sub-nanomolar range (Fig 6A).“ The authors must mean sub-micromolar range, which is what the data show.

F6C and F7A-D data should be plotted as dose response curves not a bar graph.

There is no integration of the two (three if you count morphology) endpoints of the approach. Is final cell number (implied from a metabolic assay) or the determined growth rate (from imaged cell confluence) advantageous over the other for detecting “hits” or is using the two provide more sensitivity or selectivity for “hits”. The authors reference a manuscript in preparation of a screen of over 500 compounds, so this manuscript seems to be a companion methods paper. The challenge is that the screen data in that manuscript in preparation may be important to understand any advantages of the approach reported here.

Only single drug effects are reported. In cancer therapy, drug combinations are clearly needed. What advantages do the combined readouts provide for drug combination screens?

Reviewer #3: Line 170 and 171 say that "BPH-1 cells cultured in media supplemented with 2% FBS and seeded at 500 cells/well represented the optimal assay conditions". But in the figure 4, BPH-1 cells are shown to have cell seeding density at 200 cells/well. PLease explain this confusing numbers.

Also, the authors did not clearly explain the use of AUC and AOC in their assays. They need to explain what were these calculations used to evaluate in terms of cell viability or growth or inhibition.

Reviewer #4: While not particularly novel, this is a good protocol paper for drug screens in prostate cell lines - which are notoriously difficult to work with. It has merit for publication, despite the limited amount of screen data presented. These data regarding the example drugs would be nice to include for all cell lines.

7. PLOS authors have the option to publish the peer review history of their article (what does this mean?). If published, this will include your full peer review and any attached files.

Reviewer #1: No

Reviewer #2: No

Reviewer #3: **Yes: **Manjula Nakka

Reviewer #4: No

---

## [Author Response · Author response to Decision Letter 0]

24 May 2023

PONE-D-23-06181

Automation, live-cell imaging, and endpoint cell viability for prostate cancer drug screens

PLOS ONE

We have addressed all reviewer comments in the attached file "Response to Reviewers F"

---

## [Editor Report · Decision Letter 1]

31 May 2023

Automation, live-cell imaging, and endpoint cell viability for prostate cancer drug screens

PONE-D-23-06181R1

Dear Dr. Burnstein,

We’re pleased to inform you that your manuscript has been judged scientifically suitable for publication and will be formally accepted for publication once it meets all outstanding technical requirements.

Kind regards,

Irina U Agoulnik, Ph.D.

Academic Editor

PLOS ONE
---

## [Editor Report · Acceptance letter]

5 Jun 2023

PONE-D-23-06181R1 

Automation, live-cell imaging, and endpoint cell viability for prostate cancer drug screens 

Dear Dr. Burnstein:

I'm pleased to inform you that your manuscript has been deemed suitable for publication in PLOS ONE. Congratulations! Your manuscript is now with our production department. 

Kind regards, 

on behalf of

Dr. Irina U Agoulnik 

Academic Editor

PLOS ONE